# Advances in Self-Powered Ultraviolet Photodetectors Based on P-N Heterojunction Low-Dimensional Nanostructures

**DOI:** 10.3390/nano12060910

**Published:** 2022-03-10

**Authors:** Haowei Lin, Ao Jiang, Shibo Xing, Lun Li, Wenxi Cheng, Jinling Li, Wei Miao, Xuefei Zhou, Li Tian

**Affiliations:** 1School of Materials Science and Engineering, Henan University of Technology, Zhengzhou 450001, China; jascience@163.com (A.J.); xingshibo_123@163.com (S.X.); lexie_lee0828@stu.haut.edu.cn (L.L.); wenxi_cheng@haut.edu.cn (W.C.); jinling_li@haut.edu.cn (J.L.); wei_miao@haut.edu.cn (W.M.); xuefei_zhou@haut.edu.cn (X.Z.); tianli@haut.edu.cn (L.T.); 2Henan International Joint Laboratory of Nano-Photoelectric Magnetic Materials, Henan University of Technology, Zhengzhou 450001, China

**Keywords:** self-powered, ultraviolet photodetectors, p-n heterojunction, dimensional nanostructures

## Abstract

Self-powered ultraviolet (UV) photodetectors have attracted considerable attention in recent years because of their vast applications in the military and civil fields. Among them, self-powered UV photodetectors based on p-n heterojunction low-dimensional nanostructures are a very attractive research field due to combining the advantages of low-dimensional semiconductor nanostructures (such as large specific surface area, excellent carrier transmission channel, and larger photoconductive gain) with the feature of working independently without an external power source. In this review, a selection of recent developments focused on improving the performance of self-powered UV photodetectors based on p-n heterojunction low-dimensional nanostructures from different aspects are summarized. It is expected that more novel, dexterous, and intelligent photodetectors will be developed as soon as possible on the basis of these works.

## 1. Introduction

An ultraviolet (UV) photodetector is a kind of photo-electronic device that converts a light signal to an electrical signal by using the photoelectric effect of semiconductors. As an important device for both military and civil use, ultraviolet (UV) photodetectors have attracted much attention in recent years because of their vast applications in military early warning, communication sensing, environmental monitoring, disease diagnosis, and so on [1,2,3,4,5,6,7,8,9,10]. Because UV radiation is generally divided into three bands (UVA: 320–400 nm, UVB: 280–320 nm, and UVC: 10–280 nm) according to the standards of the International Commission on Illumination [11], over the past two decades, wide-bandgap inorganic semiconductors (such as ZnO, TiO_2_, SnO_2_, ZnS, GaN, Ga_2_O_3_, etc.) have been widely used to manufacture UV photodetectors as the main components [12,13,14,15,16,17,18,19]. These UV photodetectors are mainly divided into the following two categories according to the working principle: one type uses the photoconductive effect of semiconductors; the other type works by the photovoltaic effect of heterojunction. Most of these conventional UV photodetectors need an external bias voltage to provide a driving force, which means that an external power source is required. These external power sources not only increase the circuit design complexity and processing cost of UV photodetection devices, but also limit their use in special environments and conditions. Therefore, the construction of self-powered UV photodetectors without an external power source has become a particularly important and attractive research direction in the field of photoelectric detection.

In fact, the function of the power source is to provide a potential difference for the UV photodetection system as a driving force to inhibit the recombination of photogenerated electrons and holes so as to control the moving direction of photogenerated electrons and holes [1,2,20,21,22]. If an internal potential difference can be automatically formed in the UV photodetection system, the device can work independently without an external power source. Therefore, the key problem to be solved is how to automatically form an internal potential difference in the UV detection system. Using the built-in p-n junction is an effective way to realize the independent operation of the UV photodetector without an external power source. When p-type and n-type semiconductors contact each other, a thin depletion layer will be formed at the interface of the p-n junction, which forms a built-in electric field from the n region to the p region (Figure 1) [23,24,25,26,27,28,29,30].

In recent years, a large number of self-powered UV photodetectors based on the p-n heterojunction have been reported. It is obvious that gratifying progress has been made in material synthesis, device construction, and detector performance. However, compared with the current commercial UV photodetectors, the main performance of the self-powered UV photodetectors (such as responsivity, response time, quantum efficiency, etc.) needs to be further improved [31,32,33,34,35]. In this paper, some frontier research progress in the material design and device testing of the different aspects of the p-n heterojunction low-dimensional nanostructures to improve the self-powered UV photodetector performance in recent years are summarized and analyzed. Finally, the possible opportunities and challenges in this field are prospected.

## 2. Improving the Performance of Self-Powered UV Photodetectors Based on p-n Heterojunction Low-Dimensional Nanostructures from Different Aspects

### 2.1. Construct Low-Dimensional Semiconductor Nanostructures

In the field of photoelectric detection, semiconductor materials are mainly composed of large bulk states, film states, and nano states in scientific research and practical applications. Generally speaking, their excellent properties are determined by their crystal structure, size and dimension, surface structure, and band structure [36]. Self-powered UV photodetectors based on low-dimensional semiconductor nanostructures usually show high responsivity and response speed, which is closely related to the advantages of low-dimensional semiconductor nanostructures, such as large specific surface area, excellent carrier transmission channel, and photoconductive gain much larger than that of bulk materials [37,38,39].

Shan’s group compares and studies the self-powered UV photodetectors based on p-n heterojunctions with different sizes and dimensions. The p-GaN film/n-ZnO film heterojunction device shown in Figure 2a is a typical layered structure self-powered UV photodetector [40]. Undoped n-ZnO films are deposited onto p-GaN with a thickness of about 2 μm to form a p-n heterojunction using a plasma-assisted molecular beam epitaxy (MBE) technique, and the current-voltage curve of the p-n heterojunction shows obvious rectifying behaviors. A photodetector is fabricated from this p-n heterojunction. Under back-illumination conditions, the GaN layer on one hand acts as a p-type counterpart to the n-ZnO layer and, on the other hand, as a “filter” that is transparent to the illumination light with a wavelength longer than 360 nm. It is noteworthy that this photodetector shows a narrow band photoresponse spectrum centered at 374 nm with a full width at half-maximum of only 17 nm and a peak responsivity of about 1 μA/W under back-illumination at 0 V bias due to the GaN “filter,” revealing that the responsivity of the photodetector is highly selective.

Compared with self-powered UV photodetectors based on 2D films, those based on 0D and 1D materials with large surface area to volume ratio usually show higher responsivity and shorter response time because more light can be absorbed by light capture or anti-reflection effect [41]. As shown in Figure 2b, a heterojunction UV photodiode consisting of epitaxially grown p-GaN layers and polyvinyl alcohol (PVA) coated ZnO colloidal nanoparticles exhibits a lowpass and bandpass alternative property depending on the illumination direction [42]. Under the incident optical power of 120 nW, the peak responsivity at 350 nm is about 0.225 mA/W at zero bias, indicating two orders of improvement compared to that of the previous p-GaN film/n-ZnO film heterojunction UV photodetector shown in Figure 2a. It is closely related to the fact that negatively charged oxygen ions on the surface of PVA-ZnO nanoparticles neutralize the photogenerated holes and leave more conduction-band electrons, resulting in an increase in conduction electrons. In addition, the time response of the p-GaN/PVA-ZnO NPs heterojunction shows the fastest response with a 25 ms rising time and a 50 ms falling time at 0V bias under back illumination, which demonstrates three orders of improvement compared to a photoconductor and one order of improvement compared to the MSM photodetector based on the PVA-ZnO NPs. In comparison to a p-GaN, ZnO nanowire photodiode, about a 3 order of magnitude improvement is demonstrated.

However, compared with the p-n heterostructure composed of nanoparticles and thin films, the 1D core–shell p-n heterojunction nanostructure can exhibit a higher self-powered potential due to its wider active area for light absorption and excellent carrier transmission channel. As shown in Figure 2c, a highly spectrum-selective self-powered UV photodetector has been fabricated from n-ZnO/p-NiO core–shell nanowire arrays [43]. A NiO shell layer with a thickness of about 45 nm is deposited onto the highly oriented vertical ZnO nanowires with a diameter of about 30 nm by the chemical vapor deposition technique and magnetron sputtering technique. The linear I-V curves for both Au on NiO and In on ZnO reveal good ohmic contacts, confirming that the rectification behavior arises from the n-ZnO/p-NiO interface. The outer-layer of the p-NiO acts as a “filter” which can filter out the photons with short wavelengths, making this photodetector only respond to a narrow spectrum range. It is noteworthy that the peak responsivity of the n-ZnO/p-NiO core–shell nanowire arrays photodetector at 0V bias is about 0.493 mA/W. Meanwhile, it is over twice that of a p-GaN/n-ZnO nanoparticle wavelength-selective photodiode (0.225 mA/W) and at least two orders of magnitude larger than that of an epitaxially grown p-GaN/n-ZnO heterojunction for selective UV detection (about 1 μA/W). Furthermore, the response time of the n-ZnO/p-NiO core–shell nanowire arrays photodetector is about 1.38 ms, which is estimated as the 10–90% rise time measured at zero bias. The response speed is significantly faster compared with the photoconductive photodetector, of which the decay time is usually in the order of several seconds due to the presence of deep level traps and the adsorption of gas molecules. In addition, because conductive polymers can improve the photogenerated e-h pair separation ability of inorganic semiconductors and have excellent hole transport properties [44,45,46,47,48,49,50,51,52,53], they are also widely used to construct self-powered UV photodetectors based on p-n heterojunction with inorganic semiconductors [54,55,56]. Yang’s group demonstrates a self-powered ultraviolet (UV) photodetector based on p-P3HT/n-ZnO nanowire array heterojunction, which is fabricated by spin-coating with a layer of P3HT on ZnO nanowire arrays [57]. At zero bias, this photodetector exhibits a responsivity of 0.125 mA/W and a specific detectivity of 3.7 × 10^7^ Jones, and the response time is only 90 ms and the recovery time is ≈98 ms at a low light intensity of 0.84 mW/cm^2^ for λ = 365 nm, which is much shorter than previous ZnO photodetectors (approximately in seconds). The fast response and recovery speed may be attributed to the well-defined transport path of 1D nanowires for the photogenerated carriers. Although considerable progress has been made on constructing low-dimensional semiconductor nanostructures used for self-powered p-n heterojunction UV photodetectors have been realized, there are still many limitations affecting their wide applications. One of the urgent problems to be solved is developing low-cost fabrication methods for high-quality active materials.

### 2.2. Expand the p-n Heterojunction Interface

An inner potential difference is necessary for the separation of photogenerated e-h pairs and can be automatically created due to the Fermi energy difference between two dissimilar semiconductors. Subsequently, the built-in electric field induced by the potential difference in the p-n heterojunction will guide the movement of photogenerated carriers and give rise to photocurrent [58]. However, the interface of p-n heterojunction provides an ideal place for the separation of photogenerated e-h pairs. The expansion of the heterogeneous interface is conducive to the separation of photogenerated e-h pairs, resulting in greater photocurrent. A core–shell heterostructure is a typical structure with a large heterogeneous interface that is widely used in the field of photoelectric detection [59,60,61,62,63,64,65,66,67,68,69].

Numerous self-powered UV photodetectors based on core–shell p-n heterojunction have already been reported by many groups [70,71,72,73,74]. As shown in Figure 3a, Dunn and coworkers fabricated a self-powered ZnO-Nanorod/CuSCN UV photodetector exhibiting rapid response using the method of an array of ZnO nanorods 2–3 μm in length and 70–100 nm in diameter coated with a layer of CuSCN [75]. The device performs as a self-powered UV detector that operates at a nominal zero-applied field with a photocurrent response of 4.5 μA for a low UV irradiance of 6.0 mW/cm^2^. A fast 500 ns rise and a 6.7 μs decay time are recorded with a UV-vis rejection ratio of ≈102. As a binary UV photodetector, working at an applied positive bias of 0.1 mV, a rapid detection time of 4 ns is possible. Furthermore, the responsivity increases with illumination intensity up to 7.5 mA/W with a corresponding gain of ≈0.04 for 6 mW/cm^2^ irradiance, and the device is operated at higher applied fields, and a responsivity of 9.5 A/W at −5 V is obtained.

In response to the growing need for wearable health monitoring systems, which call for a high-performance real-time UV sensor to prevent skin diseases caused by excess UV exposure, Fang’s group developed a novel high-performance self-powered UV photodetector based on p-CuZnS/n-TiO_2_ core–shell heterojunction (Figure 3b) [76]. The photodetector exhibits a responsivity of 2.54 mA/W at 0 V toward 300 nm, and the response time of the fiber-shaped photodetector is fast (both the rise and decay time are less than 0.2 s at 0 V under 320 nm). Moreover, by effectively replacing the Ti foil with a thin Ti wire for the anodization process, the conventional planar rigid device is artfully turned into a fiber-shaped flexible and wearable one. The fiber-shaped device shows an excellent responsivity of 640 A/W, an external quantum efficiency of 2.3 × 10^5^%, and a photocurrent of ≈4 mA at 3 V. These outstanding performances exceed those of most current UV photodetectors. These excellent performances are attributed to the large p-n heterogeneous interface produced by the core–shell structure, which produces excellent rectification effects and photoelectric response.

As shown in Figure 3c, Vomiero’s group prepared a self-powered photodetector based on core–shell ZnO-Co_3_O_4_ p-n heterojunction nanowire arrays [77]. Ultrathin Co_3_O_4_ films (in the range of 1–15 nm) are sputter deposited on hydrothermally grown ZnO nanowire arrays. Interestingly, a thin layer of Al_2_O_3_ buffer layer between ZnO and Co_3_O_4_ may inhibit charge recombination, boosting device performance. The photoresponse of the ZnO-Al_2_O_3_-Co_3_O_4_ system at 0 V bias is six times higher compared to the ZnO-Co_3_O_4_. The responsivity and specific detectivity of the best device are 21.8 mA/W and 4.12 × 10^12^ Jones, respectively. This could be due to the physical barrier between the p-Co_3_O_4_ and the n-ZnO layer, which reduces the recombination rate of photogenerated electrons and holes, thus physically isolating the electrons and holes during their collection at the electrodes. In addition, the alumina barrier may improve the electric field in the depletion region, helping the photogenerated e-h pairs to be separated more efficiently.

### 2.3. Improve the Crystalline Quality of Semiconductors

A large number of microscopic material units (atoms, ions, molecules, etc.) are orderly arranged to form the crystals. Compared with organic semiconductors, inorganic semiconductors are easier to form crystalline structures. In a semiconductor with low crystalline quality, there are more defects and grain boundaries, which is not conducive to the separation and transmission of photogenerated carriers. Therefore, improving the crystalline quality of semiconductors is one of the most effective ways to build high-performance self-powered UV detectors [78,79,80,81,82].

As shown in Figure 4a, Zhang’s group constructed a high-sensitivity, broadband photodetector based on 2,4-bis[4-(*N*,*N*-dimethylamino)phenyl]squaraine (SQ) nanowire/crystalline Si (c-Si) p-n heterojunctions [83]. Owing to the high crystal quality of the SQ nanowires, the heterojunctions exhibit excellent diode characteristics and the ability to detect broadband light spanning from ultraviolet (UV) light, to visible (Vis) light, to near-infrared (NIR) light. Under the 0.66 mW/cm^2^ 254 nm and 302 nm light illumination, the device exhibits a photosensitivity (defined as (I_ligh_t − I_dark_)/I_dark_) as high as 430% and a rise time of 0.6 s at a bias voltage of −3 V and reveals a power conversion efficiency (PCE) of up to 1.17%. This result also proves the potential of the device as a self-powered photodetector operating at zero external bias voltage. As shown in Figure 4b, Fang’s group fabricated a high-responsivity self-powered solar-blind UV photodetector with a high rejection ratio based on PEDOT:PSS/Ga_2_O_3_ organic/inorganic p-n junction [84]. At zero bias, the device exhibits ultrahigh responsivity of 2.6 A/W at 245 nm with a sharp cutoff wavelength of 255 nm, which is much larger than that of previous solar-blind DUV photodetectors. Moreover, the device exhibited an ultrahigh solar-blind/UV rejection ratio (*R*_245 nm_/*R*_280 nm_) of 10^3^, which is two orders of magnitude larger than the average value ever reported in Ga_2_O_3_-based solar-blind photodetectors. In addition, the photodetector shows a rise time of about 0.3 ms from 10% to 90% and a decay time of about 3 ms from 90% to 10% at −10 V.

### 2.4. Utilize Transfer of Photogenerated Carriers or Enhance the Built-In Electric Field

Under ultraviolet (UV) irradiation, inorganic semiconductors with a wide band gap show excellent photoelectric response properties, which makes them the main candidates for UV detectors. However, the recombination of photogenerated electrons and holes is a headache because it reduces the effect of converting light absorption into photocurrent, which limits the further improvement of UV photodetector performance. Therefore, how to effectively inhibit the recombination of photogenerated electrons and holes or promote the separation of photogenerated e-h pairs has become an important improvement direction [85,86].

On the one hand, the transfer of photogenerated carriers between different components is usually used to inhibit the recombination of photogenerated electrons and holes. As shown in Figure 5a, Basak’s group has developed an unprecedented high performance dual wavelength self-powered ZnO@CdS/PEDOT:PSS core–shell nanorod array photodetector through a simple aqueous chemical method wherein a suitable band alignment between ZnO and CdS has been utilized [87]. Due to the good energy level matching between ZnO and CdS, the photogenerated holes in the valence band of ZnO are transferred to CdS under ultraviolet light, and then input to the electrode through PEDOT:PSS, and while the photogenerated electrons in the conduction band of ZnO are easily transferred to the electrode, which promotes the rapid separation of photogenerated electron hole pairs and inhibits recombination, so that the photocurrent is about three times higher than that of CdS/PEDOT:PSS, reaches 2.2 × 10^−6^ A at 0 V bias (light intensity of 1 mW/cm^2^). Furthermore, temporal responses faster than 20 ms can be achieved in these solution-processed photodetectors. On the other hand, the separation efficiency of photogenerated carriers is improved by enhancing the built-in electric field. As shown in Figure 5b, Liu’s group has fabricated a super-high performance self-powered UV photodetector based on GaN/Sn:Ga_2_O_3_ p-n junction by depositing a Sn-doped n-type Ga_2_O_3_ thin film onto a p-type GaN thick film. At 0 V bias, the device shows an ultra-high response rate of 3.05 A/w to 254 nm ultraviolet light, a high UV/visible rejection ratio of *R*_254nm_/*R*_400nm_ = 5.9 × 10^3^, and an ideal detectivity of 1.69 × 10^13^ cm·Hz^1/2^·W^−1^, which is well beyond the level of previous self-powered UV photodetectors. Moreover, this device also has a low dark current (1.8 × 10^−11^ A), a high Iphoto/Idark ratio (~10^4^), and a fast photoresponse time of 18 ms. These excellent properties are attributed to making the Fermi level move close to the conduction band by doping the tetravalent element Sn in Ga_2_O_3_, which makes the built-in potential barrier in the GaN/Sn: Ga_2_O_3_ p-n structure film larger and can separate photogenerated e-h pairs more effectively in the interface depletion region of the GaN/Sn: Ga_2_O_3_ p-n heterojunction [88].

### 2.5. Adjust the Bandgap of Semiconductors for Deep UV Detection

In recent years, biological and medical research has shown the very important role of the UV-A (400–320 nm) and UV-B (320–280 nm) bands on the Earth’s biosystem and human health (erythema, eye cataracts, skin cancer, etc.). Thus, it is of great research value to develop wide-bandgap semiconductor photodetectors tailored to the various solar UV bands and blind to the visible and IR emissions [89,90]. Although a variety of semiconductor photodetectors are able to respond to UV photons, it has been recognized that GaN-based photodetectors are excellent candidates for the detection of UV radiation, and moreover, AlGaN-based photodetectors with threshold energies of 3.4 eV (GaN) up to 6.2 eV (AlN) can be fabricated by a proper choice of the Al mole fraction, which provides a good opportunity for deep UV detection [91,92,93,94].

As shown in Figure 6a, Song and coworkers fabricated a heterojunction UV photodetector based on p-CuZnS and n-GaN using a facile chemical bath deposition method [95]. Benefiting from the high hole mobility of p-type inorganic semiconductor CuZnS and the excellent electron mobility of n-type inorganic semiconductor GaN, the CuZnS/GaN heterojunction film device shows a significantly enhanced photocurrent and good rectifying behavior at 3 V and 350 nm. More importantly, the p-CuZnS/n-GaN device r presents a high photocurrent (19 mA), high responsivity (0.36 A/W), a fast response speed (0.14 ms/40 ms), and an ultrahigh on/off ratio (3 × 10^8^) under 350 nm light illumination at zero bias. In particular, the self-powered CuZnS/GaN photodetector demonstrates high detectivity (8 × 10^13^) and an ultrahigh linear dynamic range (137 dB). These excellent results reveal that the p-CuZnS/n-GaN heterojunction device can act as a high-performance self-powered UV photodetector with the prospect of wide application. Trapped in low light absorption and the quick recombination of photo-generated electron-hole pairs, the performance of available GaN-based UV photodetectors is still unsuitable for real applications. Fabricating nanoporous GaN (porous-GaN) is a promising approach to improve the light absorption due to large specific surface area and photo trap effect of porous structures [96]. As shown in Figure 6b, Qin’s group prepared a novel self-powered UV photodetector based on p-type cobalt phthalocyanine (CoPc)/n-type porous-GaN vertical heterojunction through a thermal vapor deposition method [97]. At 0 V bias, the device exhibits a photoresponsivity of 588 mA/W, a detectivity of 4.8 × 10^12^ Jones, and a linear dynamic range of 79.5 dB under 365 nm 9 μW/cm^2^ light illumination, which displays an upper level among the reported work. In addition, compared with the CoPc/flat-GaN-based photodetector, the photocurrent of this CoPc/porous-GaN-based photodetector has increased by about four times. These excellent results can be attributed to the nanopores on the GaN surface providing higher light absorptivity and a lower interface migration distance, which induced more electrons to take part in the excitation and recombination process. The AlGaN-based solar-blind ultraviolet detectors have attracted much attention due to their good chemical and thermal stability, and particularly their potential application in the deep ultraviolet region depending on the adjustable band gap (3.4–6.2 eV) [98,99,100]. As shown in Figure 6c, Razeghi’s group has presented a back-illuminated p-i-n structure with Si-In co-doped Al_0_._5_Ga_0_._5_N solar-blind UV photodetector grown on a high-quality crack-free AlN template [101]. At zero bias, the device exhibits an external quantum efficiency (EQE) of 80% and a responsivity of 176 mA/W at 275 nm. Furthermore, the UV/visible light rejection ratio exceeded six orders of magnitude. These results are closely related to a highly conductive Si-In co-doped Al_0_._5_Ga_0_._5_N layer, adjustment of the composition and thickness of the n-Al_0_._45_Ga_0_._55_N layer, and improvement of p-type Al_0_._38_Ga_0_._62_N doping. For AlGaN-based solar-blind ultraviolet detectors, the forbidden band can continuously change from 3.4 eV (GaN) to 6.2 eV (AlN) by changing the alloy composition, and the cutoff wavelengths can vary continuously between 365 nm and 200 nm in theory, which has great advantages over traditional inorganic semiconductors (ZnO, ZnS, TiO_2_, SnO_2_, etc.). However, a high density of dislocations and other structural defects hinders their use. In the future, improving the crystallization quality of AlGaN will still be an important way to promote the performance of AlGaN-based UV photodetectors [102,103].

## 3. Summary and Perspectives

This review article summarized several examples focused on improving the performance of self-powered UV photodetectors based on p-n heterojunction low-dimensional nanostructures from different aspects, resulting in the main performance parameters of those detectors (such as responsivity, response time, quantum efficiency, etc.) having been improved to a certain extent, but there is still a certain gap compared with the current commercial photodetector. An ideal self-powered UV photodetector should satisfy the 5S requirements of high sensitivity, high signal-to-noise ratio, high spectral selectivity, high speed, and high stability. Combined with the characteristics of semiconductor materials and low-dimensional nanostructures, the built-in electric field of the p-n junction should be fully utilized to meet the great demand for weak UV signal detection and faster and more sensitive devices.

So far, researchers have made a lot of efforts and many gratifying developments have been acquired, but self-powered UV photodetectors with higher performance are still highly desired. On the one hand, exploring novel nanostructures that can promote the separation of photogenerated e-h pairs and be prepared by simple and stable methods may still be an important direction in the future; on the other hand, with the emergence of new high-performance semiconductor materials (such as graphdiyne [104,105,106,107,108,109,110,111], h-BN [112,113,114,115,116], perovskites [117,118,119,120,121,122,123,124,125,126,127], black phosphorus [128,129,130,131,132,133,134,135], and MoS_2_ [136,137,138,139,140,141,142,143,144]), the situation of inorganic semiconductors as leading materials in self-powered photodetectors may be changed. In addition, inspired by the rise of wearable photodetectors in recent years, the new generation of photodetectors needs to be designed to be more dexterous and intelligent, which obviously puts forward new requirements for the design, size, and performance of materials and structures.

## Figures and Tables

**Figure 1 nanomaterials-12-00910-f001:**
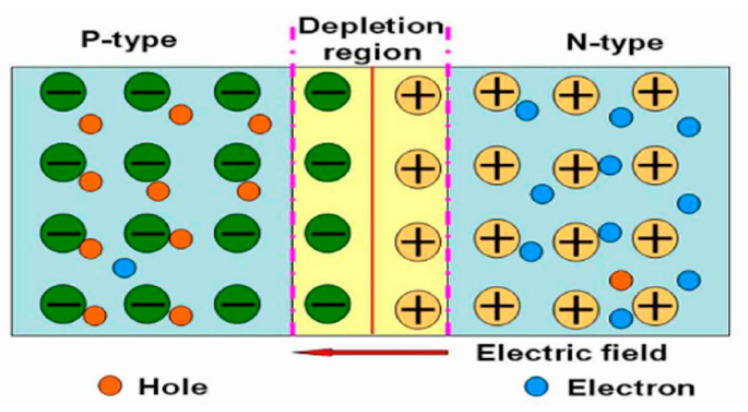
Schematic diagram of p-n junction.

**Figure 2 nanomaterials-12-00910-f002:**
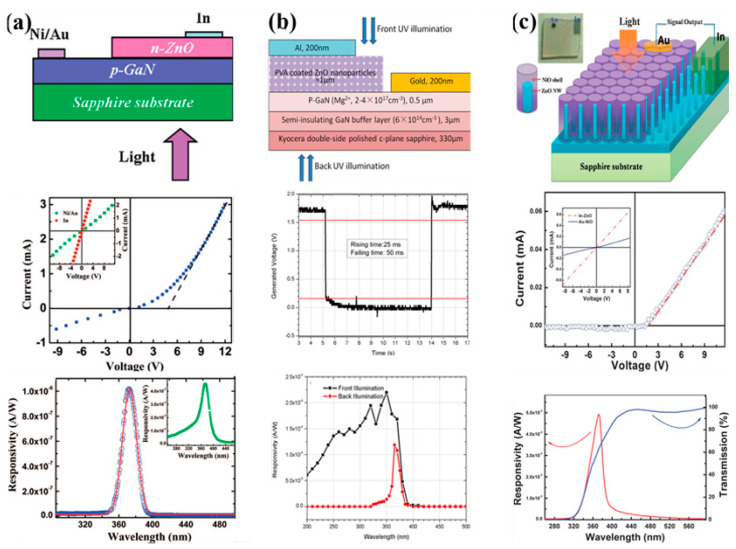
(**a**) Self-powered UV photodetector based on p-GaN film/n-ZnO film heterojunction. Reprinted with a permission from reference [40]. Copyright 2008 American Chemical Society; (**b**) self-powered UV photodetector based on p-GaN film/PVA-ZnO colloidal nanoparticles heterojunction. Reprinted with a permission from reference [42]. Copyright 2013 American Institute of Physics; (**c**) self-powered UV photodetector based on n-ZnO/p-NiO core–shell heterojunction nanowire arrays. Reprinted with a permission from reference [43]. Copyright 2013 The Royal Society of Chemistry.

**Figure 3 nanomaterials-12-00910-f003:**
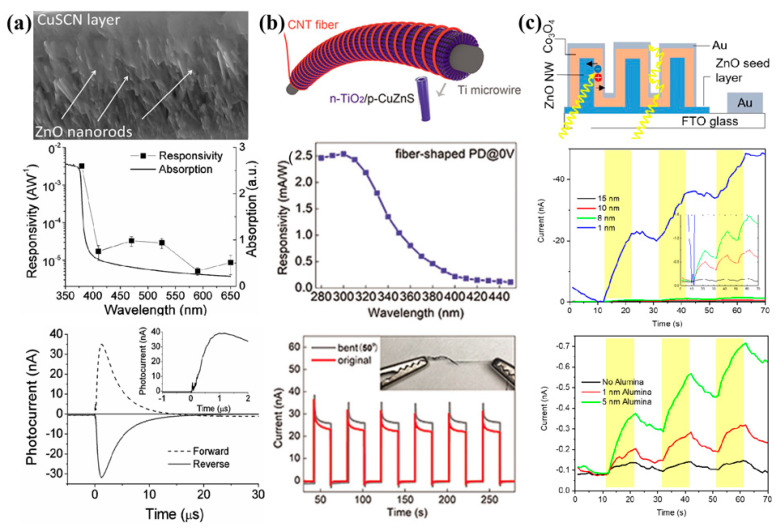
(**a**) Self-powered ZnO-Nanorod/CuSCN UV photodetector exhibiting rapid response. Reprinted with a permission from reference [75]. Copyright 2013 Wiley-VCH; (**b**) high-performance self-powered UV photodetector based on p-CuZnS/n-TiO_2_ core–shell heterojunction. Reprinted with a permission from reference [76]. Copyright 2018 Wiley-VCH; (**c**) self-powered photodetector based on core–shell ZnO-Co_3_O_4_ p-n heterojunction nanowire arrays, Reprinted with a permission from reference [77]. Copyright 2019 American Chemical Society.

**Figure 4 nanomaterials-12-00910-f004:**
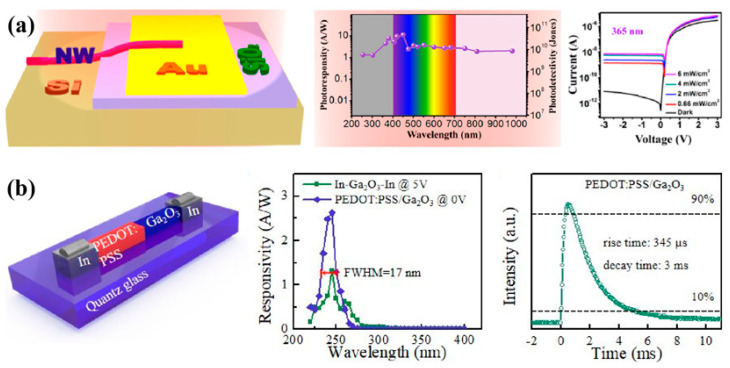
(**a**) High-sensitivity, broadband photodetectors based on organic nanowire/crystalline silicon p-n heterojunction. Reprinted with a permission from reference [83]. Copyright 2015 American Chemical Society; (**b**) high-responsivity and high rejection ratio self-powered solar blind UV photodetector based on PEDOT:PSS/β-Ga_2_O_3_ organic/inorganic p-n junction. Reprinted with a permission from reference [84]. Copyright 2019 American Chemical Society.

**Figure 5 nanomaterials-12-00910-f005:**
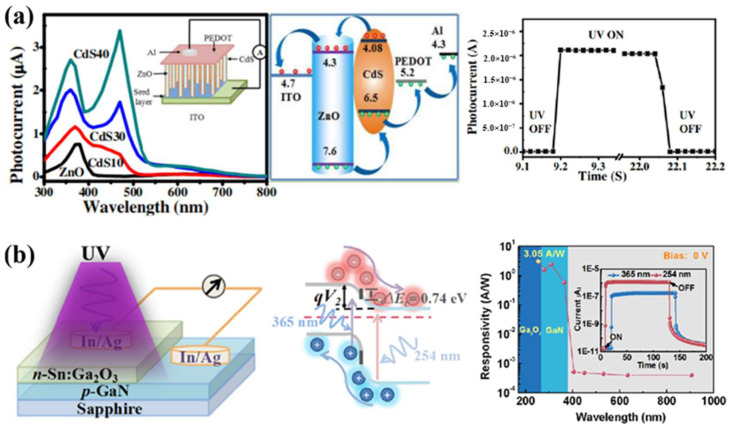
(**a**) Highly enhanced self-powered dual wavelength photodetector based on ZnO@CdS coreshell nanorod arrays. Reprinted with a permission from reference [87]. Copyright 2015 American Chemical Society; (**b**) self-powered ultraviolet photodetector with superhigh photoresponsivity based on the GaN/Sn:Ga_2_O_3_ p-n junction. Reprinted with a permission from reference [88]. Copyright 2018 American Chemical Society.

**Figure 6 nanomaterials-12-00910-f006:**
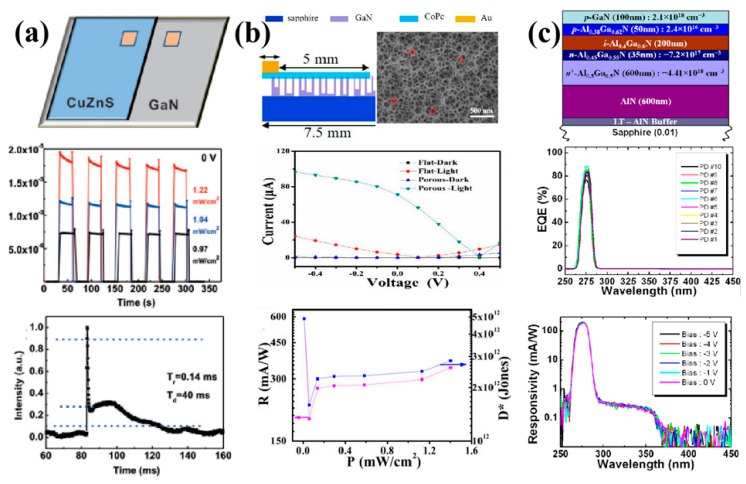
(**a**) High-performance self-powered p-CuZnS/n-GaN UV photodetectors with ultrahigh on/off ratio. Reprinted with a permission from reference [95]. Copyright 2021 The Royal Society of Chemistry; (**b**) high-performance self-powered ultraviolet photodetector based on nano-porous GaN and CoPc p–n vertical heterojunction. Reprinted with a permission from reference [97]. Copyright 2019 Multidisciplinary Digital Publishing Institute; (**c**) Al_x_Ga_1-x_N-based back-illuminated solar-blind photodetectors with external quantum efficiency. Reprinted with a permission from reference [101]. Copyright 2013 American Institute of Physics.

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
