# Peer review of "Advances in Self-Powered Ultraviolet Photodetectors Based on P-N Heterojunction Low-Dimensional Nanostructures"

_nanomaterials, 2022, doi:10.3390/nano12060910_

Round 1
Reviewer 1 Report
The authors have presented a comprehensive overview of the state of the art in UV photodetectors based on 2 P-N heterostructures, and I don't have any major concerns about the scientific content of this review article. However, before publication, I recommend extensive language editing. As it is currently written, extensive sections need to be read a few times to fully understand the meaning. I would also recommend including the following relevant citation: ACS Appl. Mater. Interfaces 2020, 12, 17, 19384–19392
Author Response
Response to Reviewer 1 Comments
Point 1: The authors have presented a comprehensive overview of the state of the art in UV photodetectors based on P-N heterostructures, and I don't have any major concerns about the scientific content of this review article. However, before publication, I recommend extensive language editing. As it is currently written, extensive sections need to be read a few times to fully understand the meaning. I would also recommend including the following relevant citation: ACS Appl. Mater. Interfaces 2020, 12, 17, 19384-19392.
Response 1: Thank you very much for your kind guidance .Due to the rush of time, we have only made a preliminary editing of the manuscript (such as line 214-218). In addition, these papers (ACS Appl. Mater. Interfaces 2020, 12, 17, 19384-19392; Nano-Micro Lett. 2017, 9, 45; Sci. Adv. 2016, 2, e1600097) have important reference value, and they have been cited in this article ([20-22] of reference).
Reviewer 2 Report
Authors reviewed the studies of the self-powered ultraviolet (UV) photodetectors (PDs) based on P-N heterojunction in the manuscript. In this review, authors discussed the ways to improve the performance of self-powered UV photodetectors based on p-n heterojunction low dimensional nanostructures. However, there still are some comments for the manuscript. The comments are listing in the list below.
- Authors should review the GaN or AlGaN based p-n junction or heterojunction UV PDs. And authors should compare the GaN or AlGaN based p-n junction or heterojunction UV PDs with other UV PDs and discuss the differences between them.
- Should the porous GaN or AlGaN UV PDs be included to the review?
3. Although authors pointed out the issues for performances of the self-powered ultraviolet (UV) photodetectors (PDs) based on P-N heterojunction low dimensional nanostructures in the subsections of the manuscript. Each issue did not have summary. Besides, authors should discuss the trade-off between issues if they have. And authors should conclude guidelines in the end of the review for the design rule of the high performance of self-powered UV
Author Response
Response to Reviewer 2 Comments
Point 1: Authors should review the GaN or AlGaN based p-n junction or heterojunction UV PDs. And authors should compare the GaN or AlGaN based p-n junction or heterojunction UV PDs with other UV PDs and discuss the differences between them.
Response 1: GaN is a superior candidate material for fabricating ultraviolet photodetectors. We added a new section (line 286-347: 2.5 Adjust the bandgap of semiconductors for deep UV detection) to the manuscript. We have summarized and analyzed some GaN or AlGaN-based p-n heterojunction ultraviolet photodetectors, and discussed the differences from other UV photodetectors.
Point 2: Should the porous GaN or AlGaN UV PDs be included to the review?
Response 2: The porous GaN-based p-n heterojunction ultraviolet photodetectors have been included to the review (line 318-330).
Point 3: Although authors pointed out the issues for performances of the self-powered ultraviolet (UV) photodetectors (PDs) based on P-N heterojunction low dimensional nanostructures in the subsections of the manuscript. Each issue did not have summary. Besides, authors should discuss the trade-off between issues if they have. And authors should conclude guidelines in the end of the review for the design rule of the high performance of self-powered UV photodetectors based on p-n heterojunction low dimensional nanostructures.
Response 3: Due to the rush of time, we have only made a preliminary editing of the manuscript, including some summaries (such as line 152-156) and guidelines (such as line 354-359).
Thank you very much for your kind guidance and help.